

# Distribution of phosphorus fractions of different plant availability in German forest soils and their relationship to common soil properties and foliar P concentrations

Jörg Niederberger[1], Martin Kohler[1], Jürgen Bauhus[1]

[1]Institute of Forest Sciences, Chair of Silviculture, University of Freiburg, Freiburg, Germany

*Correspondence to*: Jörg Niederberger (joerg.niederberger@waldbau.uni-freiburg.de)

**Abstract.** Repeated, grid-based forest soil inventories such as the nationwide German forest soil survey (GFSI) aim, among
other things, at detecting changes in soil properties and plant nutrition. In these types of inventories, the only information on
soil phosphorus (P) is commonly the total P content. However, total P content in mineral soils of forests is usually not a
meaningful variable to predict the availability of P to trees. Here we tested a modified sequential P extraction ac-cording to
Hedley to determine the distribution of different plant available P fractions in soil samples (0-5 and 10-30 cm depth) from 146
GFSI sites, capturing a wide variety of soil conditions. In addition, we analyzed relationships between these P fractions and
common soil proper-ties such as pH, texture, and organic Carbon content (SOC). Total P content among our samples ranged
from approximately 60 up to 2800 mg kg-1. The labile, moderately labile, and stable P fractions contributed to 27 %, 51 %
and 22 % of total P content, respectively, at 0-5 cm depth. At 10-30 cm depth, the labile P fractions decreased to 15 %, whereas
the stable P fractions in-creased to 30 %. These changes with depth were accompanied by a decrease in the organic P fractions.
High P contents were related with high pH-values. Whereas the labile P pool increased with decreasing pH in absolute and
relative terms, the stable P pool decreased in absolute and relative terms. Increasing SOC in soils led to significant increases
in all P pools and in total P. In sandy soils, the P content across all fractions was lower than in other soil texture types. Multiple
linear regressions indicated that P pools and P fractions were moderately well related to soil properties (r² mostly above 0.5),
and sand content of soils had the strongest influence. Foliage P concentrations in *Pinus sylvestris* were reasonably well
explained by the labile and moderately labile P pool (r² = 0.67) but not so for *Picea abies* and *Fagus sylvatica*. Foliage P
concentrations could not be related to specific P pools. Our study indicates that soil properties such as pH, C-content or soil
texture may be used to predict certain soil P pools of different plant availability, e.g. on the basis of large soil inventories, but
foliage P concentrations across tree species appear to be determined by additional variables not considered here.



# 1        Introduction

Insufficient or even critical phosphorus supply of forest trees has been repeatedly observed during the last decades. A large proportion of forest trees which were examined within the frame-work of the Second German Forest Soil Inventory (GFSI-II) showed an insufficient P nutrition (*Picea abies* L. H. Karst and *Pinus sylvestris* L. both 20 %; *Fagus sylvatica* L. 60 %) (Ilg,

2007; Wellbrock et al., 2016). Recent declines in foliar P concentrations in Scots pine, Norway spruce, Silver fir, beech and oak were also observed across Europe (Jonard et al., 2015). In contrast to agriculture, where permanent P export is compensated by fertilization, forest sites in Germany have not been P fertilized in the past to improve tree growth, with the exception of some experiments. In undisturbed forest ecosystems, P cycling is not dominated by input and output processes but by intern reallocation (transfer) processes (Newman, 1995). However, even undisturbed forest ecosystems in temperate regions develop

in the long term a negative P balance (Smil, 2000; Walker and Syers, 1976). The harvesting of forest biomass has increased in recent years and this is predicted to grow further with the increasing wood demand. Hence the export of biomass and thereby P may aggravate P malnutrition in trees (Berndes et al., 2003; Kangas and Baudin, 2003).

It has been shown that the total P content in mineral forest soils is not a significant predictor to explain the variation in the nutritional status of the different tree species, indicating that most P in soil is not or not directly plant available (Ilg et al.,

2009). Correlations between total soil P and foliage P concentrations have been observed only in a few studies, e.g. for *P. abies* (Rs 0.54) or *F. sylvatica* (Rs 0.38) (Ilg, 2007). It has been assumed that P in foliage may be more closely related to plant available soil P fractions (Ilg, 2007). However, in forests, unlike in many agricultural systems with annual crops, it has been difficult to identify a single measure or fraction of plant available P in soils that can predict tree nutritional status well. This may be related to the many mechanisms that trees developed to cope with nutrient poor soils, including mycorrhizal symbioses,

root architecture and root exudates supporting uptake of apparently unavailable P stocks and internal recycling (Fox et al., 2011; Hinsinger, 2001; Hinsinger et al., 2011; Lang et al., 2017; Schachtman et al., 1998).

Other approaches that comprise the quantification of a number of different P fractions in mineral soils have been successfully employed for forest soils to describe the potential sources of P uptake by trees. One analytical approach that allows for the partitioning of total P in mineral soils into fractions of different (plant) availability is the Hedley fractionation method (Cross

and Schlesinger, 1995; Hedley et al., 1982; Tiessen and Moir, 2008). The original method (Hedley et al., 1982), which was modified by Tiessen and Moir (Tiessen and Moir, 2008), provides all in all seven inorganic and four organic P fractions. Often these P fractions are grouped into pools of distinct plant availability. A labile, fast cycling pool (labile P), which is considered to supply the short-term P demand of plants, a slow cycling pool (moderately labile P), which can be converted into labile P forms, and a pool of occluded P (stable P), which is assumed to hardly contribute to the current plant nutrition (Guo & Yost,

1998; Stevenson & Cole, 1999; Johnson et al., 2003). There are a number of studies that have examined changes in P stocks in forest ecosystems using the Hedley fractionation method. Some of them followed the development of P fractions over time to gain information on the relevance of these P fractions for tree nutrition during eco-system development (Richter et al., 2006; De Schrijver et al., 2012). In other studies, the approach was used to investigate the influence of different forest management



systems on the distribution of P fractions in soils (Alt et al., 2011). These were case studies at single sites or at only a few different sites and thus have only a limited population of inference (Binkley and Menyailo, 2005). To our knowledge, no studies so far have addressed the distribution of P fractions and P pools in forest soils on the basis of large scale inventories. Thus, there is little information on how different soil variables such as pH-value, C and N content or soil texture, which have

been found to influence P availability (Alt et al., 2011; Franzluebbers et al., 1996; Prescott et al., 1992; Silver et al., 2000; Stevenson and Cole, 1999; Thirukkumaran and Parkinson, 2000; Turner et al., 2007) affect the distribution of different P fractions across a variety of forest soil types. Therefore, we determined the Hedley-P fractions in mineral soil samples from a subset of 145 sites of the German Forest Soil Inventory to address the following questions:

1)      How do commonly measured soil properties such as pH-value, C content or soil texture influence the distribution of P fractions of different availability?

2)      Are foliar P concentrations of forest trees related to specific fractions or pools of soil P or other soil variables?

## 2       Material and Methods

### 2.1     Sites and Samples

For the purpose of our study, we used archived soil samples from the German forest soil inventories (GFSI I and II) including samples from the state of Baden-Wuerttemberg which originated from the first GFSI in 1990, samples from the states of Hesse, Lower Saxony and Saxony-Anhalt which originated from the second GFSI in 2006. In total, 285 archived soil samples from 147 sites (Figure 1) were included in this study. From each site, if available, two samples from two depths, 0-5 cm and 10-30

cm, were analyzed. For 0-5 cm depth, 145 samples and for 10-30 cm depth 140 samples were available. The selected samples represented a wide range of total P contents (Table 1). Additional soil and site data as well as foliar element concentrations were provided by the Thünen-Institute, the Forest Research Institute Baden-Wuerttemberg, and the North-West German Forest Research Institute. In this study, we used pH-value, C and N concentration (Table 1) and the variables soil type, forest floor type, soil texture and dominant tree species. Sampling approaches and analytical methods used in the GFSI have been described

in detail by (Wolff and Riek, 1996).

**Table 1: Variation of selected soil variables for German Forest Soil Inventory samples**

**Figure 1: Location of the 147 sample sites from the German Forest Soil Inventory (GFSI) data set; B: Baden-Württemberg (n=70);**
**H: Hesse (n=23); L: Lower Saxony (n=34) and S: Saxony Anhalt (n=20).**



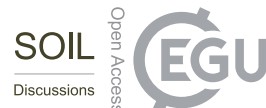

To analyze the effects of soil variables on distribution of P across fractions of different plant availability, we classified the samples by the variation in their soil properties. To classify them by pH-value, we used the buffer ranges suggested by (Ulrich, 1981). In relation to the C-content of mineral soil, samples were grouped into the classes "low" (< 1.2 % SOC), "medium" (1.2-2.8 % SOC), "high" (2.8-5.6 % SOC), and "very high" >5.6 % SOC) following the German soil assessment protocol for

5    forest soils (Ad-hoc-Arbeitsgruppe Boden, 2005). Furthermore, we used broad texture classes to group the samples. For this purpose, individual soil types of our samples were assigned to one of the following groups, sand (s), loam (l), silt (u) and clay (t), following the German soil assessment protocol for forest soils (Ad-hoc-Arbeitsgruppe Boden, 2005).

At 118 GFSI sites, P foliage concentration data for the main tree species (*F. sylvativa, P. sylvestris* or *P. abies*) were available. Foliar P concentrations for broad leaved trees were measured at leaves of the current year, for conifers, the youngest needles

10    from the most recent whorl were collected at the same time when soil samples were collected

## 2.2     Phosphorus Fractionation

The P fractionation was done using the Hedley method (Hedley et al., 1982) modified by Tiessen and Moir (Tiessen and Moir, 2008). For this purpose, 0.5 g soil was repeatedly extracted by different extractants with increasing chemical strength (Figure

15    2). A detailed description of the fractionation procedure used in this study is provided in Niederberger et al. (2015). Additional modifications of the published prescriptions like bulking fractions are described in detail in Niederberger et al. (2016).

**Figure 2: Sequential P fractionation schema according to Hedley modified by Tiessen and Moir (2008); grey boxes indicates fractions with organic and inorganic P forms; dashed line separates P pools of different availability, after Niederberger et al., (2015).**

## 2.3     Total Phosphorus

In addition to the individual P fractions, total P content for each soil sample was quantified following a nitric acid digestion. The measured total P content served also as a control to verify the recovery rate for the sum of P fractions; see also Niederberger et al. (2015)

## 2.4     Statistical analyses

In a first step, data were analyzed using descriptive statistics. Since most of our soil and site data were not normally distributed (Shapiro-Wilk-Test, p<0.05), we used non-parametric tests to identify significant differences in P pools or P fractions subject to soil properties.





To compare results between soil depths, we used the paired non-parametric Wilcoxon test. To compare the results among different soil classes within one depth, for instance classes of different soil C content, we used the non-parametric Mann-Whitney-U-test.

Additionally, we used linear regression models to explain P pools and P fractions by soil proper-ties depth, pH, SOC, and soil

type. For modelling the soil P content, we applied a log transformation to the P fractions and Pools. Transformations of the major soil types sand, loam, silt and clay into grain size distribution (expressed as the sand content) led to considerable improvements of model quality. Since only data for soil type were available in our project, we used the mean values of sand content of specific soil types based on the German soil assessment protocol for forest soils (Ad-hoc-Arbeitsgruppe Boden, 2005).

Furthermore, we used linear regression models to test whether the P nutrition status in leaves and needles of *F. sylvatica, P. abies,* and *P. sylvestris* can be explained by typical soil variables such as pH, SOC, soil texture as well as the different P pools or fractions in mineral soil.

Predictor variables used for model building were checked for correlation with Pearson correlation coefficients (r<0.7), for multiple collinearity with VIF (variance inflation factor, <10) and condition number test (<30) as well as for auto correlation

(Durbin-Watson). For all regression models, we used the stepwise backward method where non-significant predictors (p > 0.05) were progressively excluded. All statistical analyses were performed with SPSS 24 (IBM, 2011).

## 3 Results

### 3.1 Total P

The total P content calculated from the sum of all Hedley P fractions ranged from 58 to nearly 2800 mg kg-1 across the 285 GFSI samples. The independently measured total P values ranged from 42 to nearly 2300 mg kg-1 and were closely related to Hedley P sums (Spearman rs = 0.98, α<0.001). The difference between the measured mean values of total P (547 mg kg-1) and the calculated Hedley total P (563.4 mg kg-1) was approximately 3 % and not statistically significant (Wilcoxon paired rank sum test, α=0.05) (Niederberger et al., 2015).

### 3.2 Distribution of P fractions with soil depth

The sum of stable P fractions (P HCl$_{conc}$ and Pi residual) increased with increasing depth, where-as the total P content (as sum of all Hedley fractions) at 10-30 cm depth was lower than at 0-5 cm depth (Table 2). Relative shares of the stable pool related to total P increased with depth (from 22 % to 30 %) while the portion of labile P pool decreased (from 27 % to 15 %). The portion moderately-labile P pool, although decreasing significantly in absolute values (Figure S1), showed no distinct change

with depth.



**Table 2: Mean values (± standard deviation) of Hedley-P fractions for all soil samples separated by depth (all values in mg kg⁻¹) and relative shares of P pools to the sum of all Hedley P fractions.**

Organic P forms were the largest single fractions for the labile as well as the moderately labile P pools at both depths. At 0-5 cm depth the organic P forms contributed even more than the sum of the inorganic forms to these two P pools (Table 2). At 0-5 cm depth, concentrations of all labile P fractions were significantly higher than at 10-30 cm depth. While organic P in the moderately labile pool (Po NaOH) decreased with soil depth, this was not the case for inorganic P in this pool (Pi NaOH and Pi 1M HCl). The stable fractions P HCl$_{conc}$ and residual Pi showed no significant change with soil depth.

### 3.3 The influence of soil pH on distribution of P fractions

**Figure 3: Total soil P and P in pools of different plant availability (means), grouped by pH-classes and soil depths; lower case letters indicate significant differences between pH-classes within P pools and per depth (non-parametric Mann-Whitney-U-test, α < 0.05).**

The total P content decreased with increasing soil acidity (Figure 3). This decrease was mainly attributable to a significant decline in the pool of stable P fractions, which decreased both in absolute as well as relative terms. The portion of stable P dropped from 48.7 % to 16.0 % of total P at 0-5 cm depth and from 56.8 % down to 26.0 % at 10-30 cm depth (Figure 3). In contrast, the labile P pool in the surface soil (0-5 cm) increased significantly in absolute as well as relative terms with increasing acidity. At 10-30 cm depth, absolute quantities of labile P remained relatively constant, whereas its relative share of total P increased with increasing acidity (Figure 3). The moderately labile P pool showed comparatively small differences between pH classes at both depths. Only at 10-30 cm depth, this pool was significantly smaller in the most acidic soils when compared to the other pH-classes.

Considering individual P-fractions, we found for all labile P fractions an increase of P concentration with decreasing pH-value at 0-5 cm depth (Table S1), but this was significant only for the Po HCO$_3$-fraction. At 10-30 cm depth, the labile P fractions also increased with decreasing pH-value up to a maximum at pH 4.2-5.0 but then declined for the most acidic soils (pH 3.0-4.2). At both soil depths, the percent contribution of the Pi NaOH fraction to total P increased from neutral to acidic conditions, except for the most acidic soil group. Whereas the size of the Po NaOH fraction was unrelated to soil pH at 0-5 cm depth, it declined in the most acidic soils at 10-30 cm depth. For the Pi 1M HCl fraction we found inconsistent but significant differences among pH classes at 0-5 cm and 10-30 cm depth. At both depths, the lowest concentration was found in the most acidic soils. The two stable P fractions P HCl$_{conc}$ and P residual showed clear and consistent significant decreases with the decreasing pH at both depths (Table S1 and S2).



### 3.4 The influence of soil organic carbon on distribution of P fractions

Via the amount of organic matter, C-content is directly linked with the (organic) P-content in mineral soils. Total P as well as P in all Hedley fractions increased strongly and significantly with increasing C-concentration in mineral soil at both depths (Fig. 4). In contrast, the relative proportions of the P pools showed no or only minor changes with increasing soil C-content at both depths (Fig. S2, Table S3 and S4).

**Figure 4: Hedley P pools and total P (mean values), grouped by C-content in % and depth; n = number of observations, lower case letters indicate significant differences in P pools among C content classes and per depth, non-parametric Mann-Whitney-U-test, α < 0.05. The column with no letters had too few observations for statistically valid tests for differences.**

### 3.5 The influence of soil texture on P fractions

Total P and in particular the stable P pool increased with the decrease of the coarse fraction in soil texture. The lowest concentrations of total P as well as P in the stable and moderately stable fractions were found in sandy soils at both depths (Fig. 5). In particular, the 1M HCl soluble P fractions showed extremely low P concentrations in sandy soils (Table S5). Accordingly, the highest relative contribution of labile P to total P was also found in sandy soils; 40 % at 0-5 cm and 35 % at 10-30 cm depth. The relative proportion of labile P decreased with decreasing grain size. In 10-30 cm depth, the increase of total P with increasingly finer soil texture was mainly caused by the stable P forms.

**Figure 5: Hedley P pools and total P (mean values) grouped by soil texture and depth, n = number of observations, lower case letters indicate significant differences in P pools among soil texture groups, non-parametric Mann-Whitney-U-test, α < 0.05.**

### 3.6 The combined influence of soil variables on Hedley soil P pools and P fractions

**Table 3: Results of linear regression models for log transformed P pools and P fractions, by soil variables in 0-5 cm soil depth, model quality and standardized regression coefficients.**

We found the highest goodness of fit in multiple linear regression models for the organic P fractions of labile and the moderately labile and stable P pool, whereas model performance for the inorganic P forms were considerably lower (Table 3). Sand content was a negative predictor in all cases, whereas SOC was always a positive predictor. In contrast, pH value was a negative predictor for labile P and in particularly labile organic P, but positive for the stable P forms.





Whereas for labile P pool and the organic P fractions the C content had a stronger influence than the sand content, the reverse was true for stable P pools and. For moderately labile P, both predictors were similarly important, except for the 1M hydrochloric acid soluble P fraction, where C content was not a significant predictor.

Models with a goodness of fit above 0.4 (Table 3) were only obtained for organic P fractions (Po HCO$_3$, Po NaOH) or P

fractions with high portions of organic compounds (P HCl$_{conc}$) and fractions which are dominated by soil parent material (P HCl$_{conc}$, P residual). As an exception, the 1M hydro-chloric acid soluble P fraction was the only inorganic P fraction with a model quality above 0.4.

In samples from 10 to 30 cm depth (Table S6), we found in general lower model qualities but comparable patterns to those observed at 0-5 cm depth.

### 3.7    The influence of soil variables and Hedley P pools and P fractions on foliar P concentrations

Total soil P as well as P pools of different availability varied considerably between GFSI plots dominated by different tree species (Table 4). Under *P. sylvestris* all soil P pools were significantly lower than under Fagus sylvatica or *P. abies*, nevertheless, yet the relative proportion of labile P was greatest under *P. sylvestris*. The differences in P pools between *P.*

*abies* and F. *sylvatica* were mostly small and only for the stable P pool significant.

**Table 4: Mean values (± standard deviation) of the concentration of total P and P in pools of different plant availability in soils of GFSI plots dominated by different tree species. Different letters indicate significant differences between tree species (non-parametric Mann-Whitney-U-test, α < 0.05).**

The quality of the linear regression models for foliage P content varied considerably amongst the examined tree species. For F. sylvatica and P. abies the adjusted r² of models did not reach 0.5. In contrast, models to explain foliar P content of P. *sylvestris,* through P pools and soil variables at 0-5 cm depth were of considerably quality. Here P pools were the most important predictors, whereas soil variables played only a minor role. Interestingly, the moderately labile P pool had a negative

influence on foliage P content in *P. sylvestris* at 0-5-cm depth. In all multiple linear regression models, at least one of the three soil P pools was found to be a significant predictor of foliar P content (Table 5), however no specific P pool was significant in all cases. With the exception of *P. abies* soil samples from 0 to 5 cm depth, P pools had a stronger and significant influence on model quality compared to soil variables in all other models.

However, the vast majority of the relatively low number of foliage samples of *P. sylvestris* originated from sites with very

similar soil properties (same humus type, soil texture, soil type and pH class) and in general very low total P concentrations (significantly lower than *P. abies* and *F. sylvatica* which did not differ significantly (Table 3)).

**Table 5: Model quality, regression constants and standardized, significant regression coefficients of linear regression models to explain foliage P content with P pools of different availability and other soil variables at 0-5 and 10-30 cm soil depth.**





Multiple linear regressions models to predict foliage P concentration in the three tree species with soil variables and P pools from 0-5 and 10-30 cm depths achieved a moderate quality when applied across the three species (Fig. 6). However, they were not suited to predict foliage P concentrations when datasets were considered for each of these species separately.

**Figure 6: Linear regressions to predict foliage P concentration in *P. abies*, *P. sylvestris*, and *F. sylvatica* (see Table 5) with soil variables from 0-5 (left panel) and from 10-30 cm depth (right panel), x-axis measured and y-axis modelled foliage P concentration in mg g⁻¹.**

## 4     Discussion

### 4.1     The influence of soil properties on Hedley P pools and fractions

Our results show that soil properties like acidity, soil C-content, soil texture and soil depth have an important influence on the quantity and distribution of plant available P in forest soils. Yet there have been very few studies that investigated this issue (Augusto et al., 2017; Buckingham et al., 2010; Shang et al., 1992; Zederer and Talkner, 2018).

In our analysis there was no single soil variable that was consistently the best predictor of the different Hedley pools or fractions. However, there were consistent patterns such as that SOC content always had a positive and sand always a negative influence, whereas pH could have both, positive and negative influences. Yet pH was never a stronger predictor than either C or sand. Labile Pi fractions could not be sufficiently well explained ($r^2 > 0.4$) with the chosen soil variables. They may be more influenced by soil biological processes, which were not captured in our study. In contrast organic P fractions were well described by org. C content. Sand content was the strongest predictor of variation in moderately labile and stable P fractions and pools. This reflects the influence of soil clay content, which is inversely related to sand content, on these fractions (e.g. Zederer and Talkner 2018). The magnitude of stable soil P pools was strongly and positively related to pH at both soil depths, whereas the moderately labile pool was, in absolute terms, little affected by soil pH.

In the following discussion, we first address the general assumptions regarding the influence of these variables on P distribution in mineral soils and relate them to our results. Secondly, we discuss the results of the regression models that were used to examine the relationship between Hedley P pools or fractions and foliar P concentration of *F. sylvatica, P. abies,* and *P. sylvestris* which are the main tree species in Germany.

### 4.1.1     Soil depth

The P distribution within soil depth is dominated by the processes of biological turnover. Plant P uptake and transformation of inorganic in organic P in combination with microbial activity lead to an enrichment in organic as well as in labile P and hence



in total P in the topsoil and forest floor layers (Jobbagy and Jackson, 2001). This could be clearly demonstrated by our results which showed significant higher amount of organic P forms in the upper soil layer, as well as labile P in total, although labile P forms did not dominate the surface soil layer but contributed on average 27%. Furthermore, our findings of significant increase in organic P fractions (Po Na-HCO$_3$, Po NaOH) with decreasing soil depth are in accordance with earlier studies

(Condron et al., 2005; Turner et al., 2003) that described a strong influence of microorganism on organic P accumulation in the topsoil layers, and the decrease of organic carbon with soil depth that strongly affected the soil Po content (Jobbágy and Jackson, 2000).

The observed decrease of total P with soil depth (23% from 5 to 30 cm depth) in our study seems to support previous studies which pointed out that commonly more than 40 % of the stored P, based on profiles of 100 cm depth, can be found in the

topmost 20 cm (Jobbagy and Jackson 2001).

A further known effect with increasing soil depth is the relative accumulation of more stable inorganic P forms in deeper soil layers by fixation of P for instance as secondary P minerals or in clay minerals (Buckingham et al., 2010; Vitousek et al., 2010). These effect could also be observed in our results, nevertheless this increase of moderately labile and stable Pi forms were, although perceivable, not significant. It might be that this effect is stronger if the depth gradient is more pronounced.

**4.1.2    Soil pH**

The solubility and fixation of soil P is strongly affected by soil acidity (pH) (Hinsinger, 2001; Shang et al., 1992; Stevenson and Cole, 1999). Optimum availability of P to plants occurs typically around pH of 6.5. Below a pH of 6, P is being fixed as Fe or Al phosphates or adsorbed by ox-ide surfaces (Shang et al., 1992; Stevenson and Cole, 1999) and above a pH of 7, P is being fixed in the form of Ca phosphates (Stevenson and Cole, 1999). It has been shown that there are considerable changes

in the relative importance of P fractions with pH even if there are only a mi-nor or no influence of pH on total P (Turner and Blackwell, 2013). For example, DNA and phosphonate were only found in most acidic soils (Turner and Blackwell, 2013). However, the distribution patterns of specific organic P forms can unfortunately not be picked up by the Hedley fractionation. In accordance with other studies, in our study the highest portion of labile P was found in soil samples with the lowest pH (Alt et al., 2011; Turner and Blackwell, 2013). The decline in labile organic P with higher pH might be caused by an enhanced

mineralization of organic P by microbes which is found in soils with high pH (Stewart and Tiessen, 1987). In contrast to Turner and Blackwell (2013), in our study we found trends from low to high total P contents with increasing pH at both soil depths. This difference between our and their study may be attributed to the much larger and more variable data set used in our study. The results of our multiple regression analysis showed no uniform effect of pH as a predictor of P pools in soil. The contrasting effects of pH, negative on labile and positive on stable P forms, might be explained by the different processes that influence P

availability and P fixation of mineral soils at different pH (Hinsinger, 2001; De Schrijver et al., 2012). The negative impact of increasing pH value on labile soil P might be caused by the enhanced decomposition of organic matter by microorganisms at higher pH and increased mixing of organic matter with the mineral soil matrix (Paré and Bernier, 1989; Scheffer and Schachtschabel, 2010). This may also partially explain the positive effect of pH on stable P together with higher portions of





clay minerals (Sugihara et al., 2012) as well as primary and or secondary P containing minerals in soils at higher pH (Hinsinger, 2001).

### 4.1.3 Organic Carbon content

Since organic P forms can count for more than 50 % of total P in mineral soils (Fox et al., 2011; Stevenson and Cole, 1999; Turner, 2008), we assumed a strong influence of C-content on the distribution of P in mineral soils. Because the C-content in mineral soils decreases significantly with depth (Jobbágy and Jackson, 2000) we expected to observe a decrease in P with increasing depth.  The distribution of soil P in our samples was closely related to C content. Total P and all P fractions increased with increasing C content whereas the relative contribution of P in pools of different availability to total P remained stable

(Fig. 4). Like other studies (Cleveland and Liptzin, 2007; Perakis et al., 2017; Zederer and Talkner, 2018) we found a strong positive correlation between SOC and total organic P content ($rS = 0.77$) in mineral soils. For the inorganic P fractions, we found in accordance with Johnson et al. (2003) an increase with rising soil C con-tent. This increase could be caused by higher microbial turnover as a source of labile Pi (Condron et al., 2005; Johnson et al., 2003) in soils with higher C content. Since P fractions in our study were derived from archived and dried samples, the increase in inorganic P, in particularly in the labile P

pool, might have been caused by released microbial P during the soil drying process (Johnson et al., 2003; Tiessen and Moir, 2008). Our regression analyses indicated that SOC is a strong positive predictor variable for the labile and the moderately labile P pool and to a reduced extent also for the stable P pool. Since organic P is strongly linked with C content respectively the amount of organic matter in mineral soils (Tyler, 2002; Wang et al., 2008), it is not surprising that we found in all cases where C content was a significant predictor a positive influence in our regression models.

The rather constant relative proportions of stable, moderately labile, and labile P pools across soils with different organic matter contents (Fig. 4) and the close relationship of SOC and P (Johnson et al., 2003) suggests that these pools may be predicted by SOC content.

### 4.1.4 Soil texture

In our study, sandy soils contained significantly less P in all Hedley fractions when compared to other soil texture types. This may be related to the lower organic matter content, the fewer possible fixation opportunities e.g. at clay minerals and the higher acidity of these soils. To our knowledge, the effect of texture on the distribution of P in forest soils was so far described only in one publication (Zederer and Talkner, 2018). In a study on the influence of soil texture on the resorption capacity after P fertilization in agricultural soils significant increases in moderately labile P forms three years after P fertilization were found

at sites with high clay content (35%), whereas at sites with low clay content (5%) no significant changes were observed (Sugihara et al., 2012). Strong negative correlations between sand content and organic P and strong positive correlations with inorganic P were found, whereas silt content had the opposite effect (Halloran et al., 1985). In forest soils of northern Germany,



strong positive effects of clay minerals on Po content were identified (Zederer and Talkner 2018). These were explained by sorption of phosphate monoesters at clay minerals. Likewise, we interpret negative effects of the proportion of sand on total P content in all significant models of our study as the decreasing amount of sur-faces to which P could be adsorbed or fixed. Since 2:1 clay minerals are able to fix P, the absence of clay minerals could lead to considerably lower P content in sandy soils

(Buckingham et al., 2010). It was shown that sand content had a direct negative influence of P content in soils of all climate zones worldwide (Augusto et al., 2017). GFSI plots under *P. sylvestris* (Table 5), growing in most cases on sandy soils, showed for labile and moderately-labile P forms the strongest influence of C content and in some cases pH, indicating a dominance of microbiological processes, whereas stable P forms were solely influenced by soil type. With increasingly finer soil texture, increased higher total P contents were observed, but this was largely attributable to in-creases in the stable fraction, whereas

there were no or only minor increases in the labile or moderately-labile pools, confirming results from earlier studies (Tiessen et al., 1983, Tiessen et al., 1984). These results show that in particular the proportion of stable P pools may be influenced and explained by soil texture.

## 4.2  Foliage P concentrations in relation to soil P pools

The mean values of P in pools of different availability in soil from forests dominated by the three different trees species (Table 4) indicated considerable species-specific differences in P availability. Whereas for *Pi. abies* and *F. sylvatica*, no significant difference in P pools were observed, except for a significantly lower stable P pool under spruce, there was significant less P in all pools under *P. sylvestris*. Based on mean stone content and typical bulk densities for these depths, the average stocks of labile P in topsoil (0-30 cm depth) in *P. abies* and *F. sylvatica* stands at the studied GFSI sites were calculated at 340 kg P ha-

1 and about 200 kg P ha-1 for *P. sylvestris*. The estimated average annual uptake demand of about 4 kg P ha-1 by *P. abies* or 6 kg P ha-1 by *F. sylvatica* (George and Marschner, 1996) would be easily matched in all cases by the P in the labile pool. Some earlier studies indicated that there was a correlation between total P content in soil and tree nutrition e.g. for *P. abies* (Rs 0.54) and *F. sylvatica* (Rs 0.38) (Ilg et al., 2009; Khanna et al., 2007). Other studies showed that P fertilization lead to a significant increase in P foliage content (Prietzel and Stetter, 2010). In our study there was no single mineral soil P pool or soil

variable that was consistently the best predictor of foliage P content in the different tree species. Total organic carbon and soil clay content were best predictors ($r^2$=0.6) to model P foliage content in *F. sylvatica* (Zederer and Talkner, 2018). The major difference between Zederer and Talkner (2018) and our study was the distinct fractionation schemes which are not, in particularly not in the matter of plant availability, directly comparable. Additionally, our dataset showed greater variability in soil types and soil chemical and physical properties.

It seems that foliar P concentrations in *F. sylvatica* and *P. abies* were influenced differently by P pools and soil properties. Foliage P concentration in F. sylvatica was predicted by SOC content and the stable as well as the labile P pool, whereas in *P. abies*, it was a function of sand content and the moderately labile P pool. The reason for this might be that the average sand content in mineral soil was twice as high under *P. abies* as under F. sylvatica. Thus, sites dominated by *F. sylvatica* had a





considerably higher portion of stable organic P forms (P HCl$_{conc}$) which could lead to a negative influence on P nutrition. The finding that SOC has an important influence on P foliage content of *F. sylvatica* corresponds with findings of other studies and further-more it indicates that the forest floor mass or thickness should be included as a variable to improve model quality (Talkner et al., 2015).

Soil pH was not a significant predictor in our regression models, as was also found in previous studies (Zederer and Talkner, 2018). Our best multiple linear regressions models for foliage P in *P. sylvestris* reached quality levels ($r^2$=0.67 at 0-5 cm and $r^2$=0.52 at 10-30 cm depth) that were comparable to the best model by Zederer and Talkner (2018) for *F. sylvatica*.

The model results for *P. sylvestris* indicated that the labile soil P pool could have a strong influence for P nutrition. Nevertheless, the less available P forms, in particularly, the moderately labile P fractions, and additionally mineral soil

properties were in other cases important variables to describe the P foliage concentrations. This indicates an important difference between forests and agricultural systems. In forests, P uptake may be from both, the labile and moderately labile P pool (Niederberger et al. 2016). The latter may also replenish the labile pool during periods of reduced plant P uptake.

Although the quality of these statistical models to predict foliage P from soil properties may not appear very high, other approaches to monitor nutrient content in plant biomass using remote sensing techniques do not appear to provide better

predictions. Using hyperspectral images of the canopy, model qualities to predict the P content in the perennial grass Holcus lanatus reached values of $r^2$=0.33 respectively 0.26 depending on the algorithm chosen (van Buul, 2017). For forest sites, the model quality was considerably lower and reached an $r^2$ of 0.269 for P (van Buul, 2017) and 0.69 for N (Loozen et al., 2018) in a Catalonian pine dominated mixed species stand.

While the possibility to predict P foliage content from soil properties may appear to be limited ($r^2$ 0.6 for *F. sylvatica* in Zederer

and Talkner (2018), for *P sylvestris* $r^2$=0.67 and $r^2$=0.52 in our study), model quality may improve, if more soil and site variables are considered (especially for the forest floor).

## 5    Conclusions

Using the Hedley fractionation approach, we assessed the distribution of soil P forms of different availability in a range of German forest soils and analyzed relationships between these fractions to selected soil properties. Although our data set was

not representative for German forest sites, it clearly showed that approximately half of the soil P is contained in moderately-labile fractions, whereas stable and labile fractions contribute to ca. one quarter of total P in the up-per mineral soil. With increasing depth, the labile pool declines in favor of the stable pool. Common soil properties such as pH, SOC and soil texture may be used to predict certain P pools in large forest soil inventories. However, additional soil and site variables should be considered to improve these models.

Despite the high quantities of P in labile fractions in mineral soils, the correlations between these fractions and foliar P concentrations of *P. abies* and *F. sylvatica* were not stronger than for moderately labile fractions. Models using soil properties and soil P pools of different plant availability are not yet adequate to explain the P nutrition status in tree foliage. However,




these models may be developed further through using a larger and more balanced data set and incorporation of other important variables such as the forest floor.

## 6    Acknowledgements

J. Niederberger was financed through a grant provided to Jürgen Bauhus by the Federal Ministry of Food, Agriculture and Consumer Protection (BMELV), administered through the Thünen-Institute, Eberswalde, Germany. We wish to thank Renate Nitschke and K. Winschu for the technical support, the Forest Research Institute Baden-Wuerttemberg in the person of Dr. K. v. Wilpert and the Northwest German Forest Research Station in the person of Dr. Jan Evers for providing the GFSI samples.

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



**Table 1:** Variation of selected soil variables for German Forest Soil Inventory samples

| | | n | Min | Max | Mean ± Stdv |
|---|---|---|---|---|---|
| 0-5 cm | pH-$H_2O$ | | 3.29 | 7.47 | 4.35 ±0.79 |
| | $C_{tot}$ [g kg$^{-1}$] | | 11.42 | 314.63 | 67.76 ±40.59 |
| | $N_{tot}$ [g kg$^{-1}$] | 145 | 0.13 | 14.20 | 4.07 ±2.29 |
| | C/N | | 10.00 | 436.72 | 19.76 ±35.18 |
| | P [g kg$^{-1}$] | | 0.09 | 2.22 | 0.61 ±0.37 |
| 10-30 cm | pH-$H_2O$ | | 3.56 | 8.20 | 4.67 ±0.76 |
| | $C_{tot}$ [g kg$^{-1}$] | | 0.21 | 115.00 | 19.29 ±17.55 |
| | $N_{tot}$ [g kg$^{-1}$] | 140 | 0.11 | 8.13 | 1.38 ±1.15 |
| | C/N | | 1.94 | 60.50 | 14.66 ±6.53 |
| | P [g kg$^{-1}$] | | 0.04 | 2.29 | 0.48 ±0.35 |



**Table 2: Mean values (± standard deviation) of Hedley-P fractions for all soil samples separated by depth (all values in mg kg$^{-1}$) and relative shares of P pools to the sum of all Hedley P fractions.**

| P fraction | P pool | 0-5 cm depth n = 139[$)] | | 10-30 cm depth n = 139[$)] | | #) |
|---|---|---|---|---|---|---|
| | | % | mg kg$^{-1}$ | % | mg kg$^{-1}$ | |
| Pi resin | | | 49.9 ±36.9 | | 21.1 ±24.7 | *** |
| Pi NaHCO$_3$ | P labile | 26.9 % | 33.7 ±40.5 | 15.3 % | 22.5 ±36.7 | *** |
| Po NaHCO$_3$ | | | 86.8 ±57.9 | | 31.7 ±29.6 | *** |
| Pi NaOH | | | 85.3 ±92.9 | | 96.7 ±106.0 | |
| Po NaOH | P moderately labile | 51.5 % | 194.2 ±136.9 | 54.9 % | 118.2 ±103.2 | *** |
| Pi 1M HCl | | | 47.2 ±95.9 | | 54.8 ±137.9 | |
| P HCl$_{conc}$ | | | 96.1 ±99.2 | | 103.5 ±110.0 | |
| Pi residual | P stable | 21.6 % | 41.2 ±30.9 | 29.8 % | 42.5 ±33.5 | |

#) paired non-parametric Wilcoxon test, *** α<0.001; $) reduced number of samples since not at all sites, both soil depths were not available.



**Table 3: Results of linear regression models for log transformed P pools and P fractions, by soil variables in 0-5 cm soil depth, model quality and standardized regression coefficients.**

| depth 0-5 cm | n = 143 | predictor variables | | |
| --- | --- | --- | --- | --- |
| | | SOC | sand | pH |
| | | mg kg$^{-1}$ | % | |
| *target* | r² # | standardized regression coefficients§ | | |
| log P labile | 0.52 | **0.518** | -0.374 | -0.346 |
| log P moderately labile | 0.59 | 0.408 | **-0.539** | |
| log P stable | 0.60 | 0.132 | **-0.566** | 0.316 |
| log pi resin | *0.35* | 0.378 | **-0.387** | -0.238 |
| log Pi HCO$_3$ | *0.22* | **0.331** | -0.262 | |
| log Po HCO$_3$ | 0.61 | **0.563** | -0.340 | -0.479 |
| log Pi NaOH | *0.35* | 0.140 | **-0.542** | |
| log Po NaOH | 0.64 | **0.505** | -0.488 | |
| log Pi 1M HCl | 0.41 | | **-0.582** | 0.134 |
| log P HCl$_{conc}$ | 0.55 | 0.146 | **-0.544** | 0.291 |
| log P residual | 0.47 | | **-0.472** | 0.364 |

# adjusted r², P pools or P fractions with model performance r² below 0.4 are shown in italics. § significant predictors (p < 0.05, F-Test) are shown, strongest predictor are presented in bold.

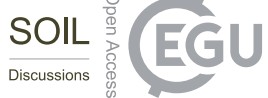



**Table 4: Mean values (± standard deviation) of the concentration of total P and P in pools of different plant availability in soils of GFSI plots dominated by different tree species. Different letters indicate significant differences between tree species (non-parametric Mann-Whitney-U-test, α < 0.05).**

| Sites | n | P total mg kg[-1] | | | P labile[#] mg kg[-1] | | | P moderately labile[#] mg kg[-1] | | | P stable[#] mg kg[-1] | | |
|---|---|---|---|---|---|---|---|---|---|---|---|---|---|
| *Fagus sylvatica* | 73 | 737.0 | ±523.0 | a | 128.9 | ±104.0 | a | 402.9 | ±374.6 | a | 205.0 | ±157.7 | a |
| *Picea abies* | 121 | 620.3 | ±426.3 | a | 134.8 | ±105.2 | a | 344.5 | ±315.4 | a | 141.0 | ±110.8 | b |
| *Pinus sylvestris* | 17 | 230.3 | ±150.4 | b | 71.7 | ±45.5 | b | 103.7 | ±78.4 | b | 55.0 | ±80.3 | c |




**Table 5: Model quality, regression constants and standardized, significant regression coefficients of linear regression models to explain foliage P content with P pools of different availability and other soil variables at 0-5 and 10-30 cm soil depth.**

| | | n | r² # | P labile | P moderately labile | P stable | SOC mg kg⁻¹ | sand (%) | pH |
|---|---|---|---|---|---|---|---|---|---|
| | | | | | standardized regression coefficients§ | | | | |
| *Picea abies* | | 61 | *0.20* | | 0.255 | | | **-0.288** | |
| *Fagus sylvatica* | 0-5 cm | 37 | *0.28* | **0.507** | | 0.354 | -0.495 | | |
| *Pinus sylvestris* | | 17 | 0.67 | **1.191** | -0.620 | | | | |
| *Picea abies* | | 59 | *0.27* | | **0.361** | | | -0.293 | |
| *Fagus sylvatica* | 10-30 cm | 35 | *0.21* | | | **0.607** | -0.487 | | |
| *Pinus sylvestris* | | 16 | 0.52 | | **0.832** | | | 0.777 | |

# adjusted r², P pools or P fractions with model performance r² below 0.4 are shown in italics, § significant predictors (p < 0.05, F-Test) are shown, strongest predictor are presented in bold.





**Figure 1: Location of the 147 sample sites from the German Forest Soil Inventory (GFSI) data set; B: Baden-Württemberg (n=70); H: Hesse (n=23); L: Lower Saxony (n=34) and S: Saxony Anhalt (n=20).**



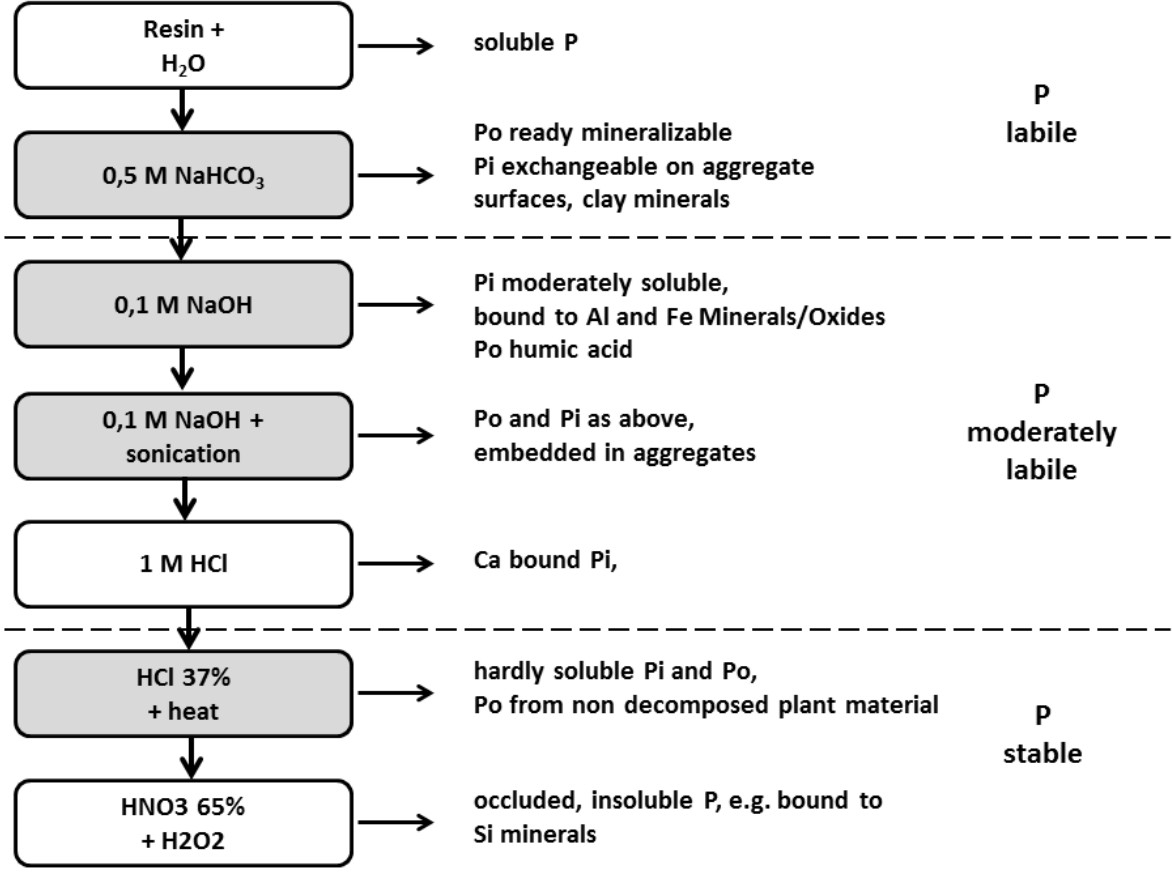

**Figure 2: Sequential P fractionation schema according to Hedley modified by Tiessen and Moir (2008) ; grey boxes indicates fractions with organic and inorganic P forms; dashed line separates P pools of different availability, after Niederberger et al., (2015).**



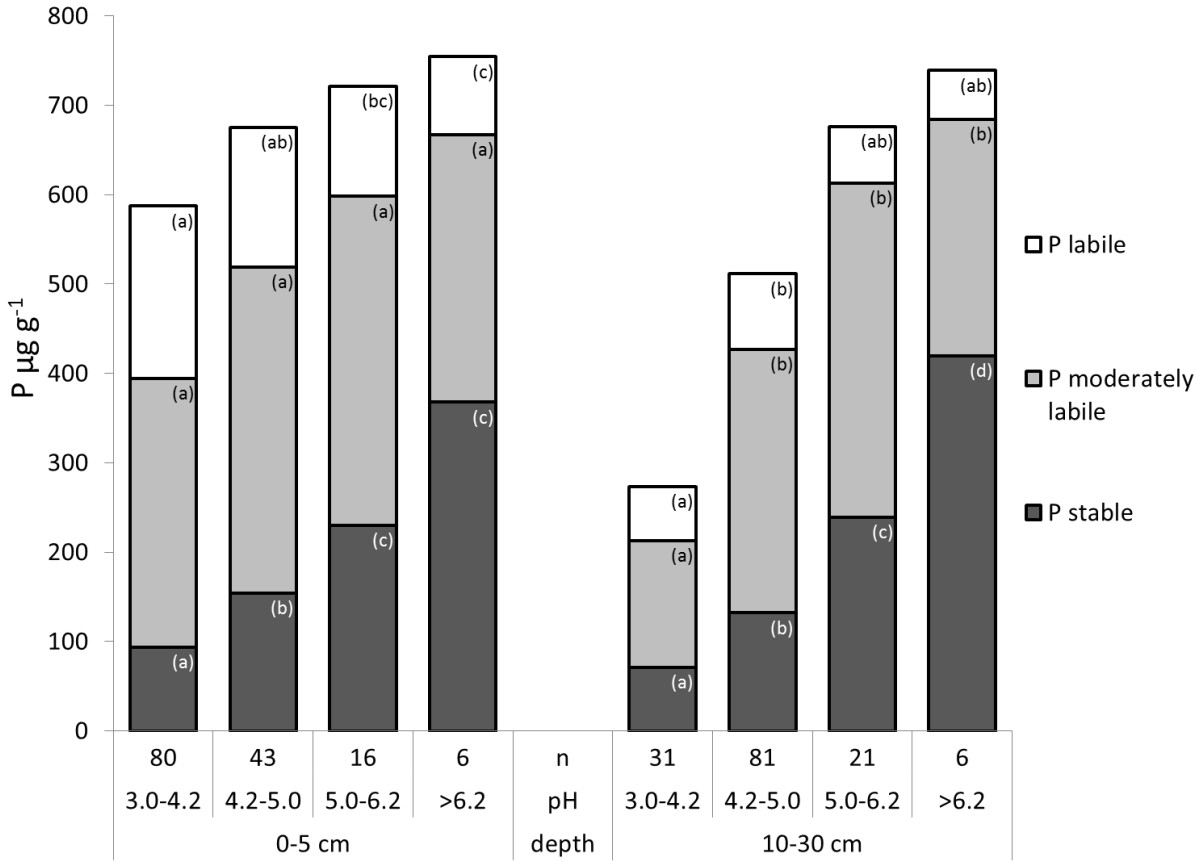

**Figure 3: Total soil P and P in pools of different plant availability (means), grouped by pH-classes and soil depths; lower case letters indicate significant differences between pH-classes within P pools and per depth (non-parametric Mann-Whitney-U-test, α < 0.05).**



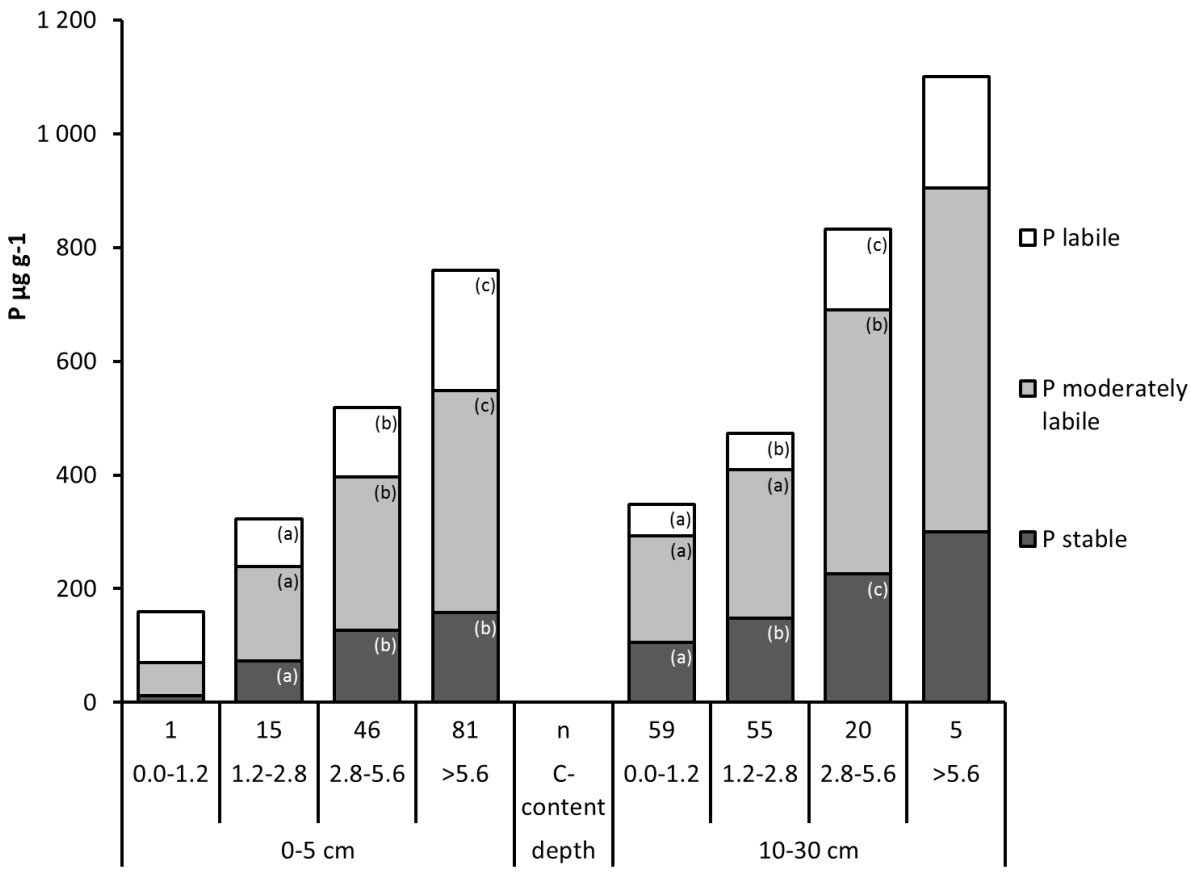

**Figure 4: Hedley P pools and total P (mean values), grouped by C-content in % and depth; n = number of observations, lower case letters indicate significant differences in P pools among C content classes and per depth, non-parametric Mann-Whitney-U-test, α < 0.05. The column with no letters had too few observations for statistically valid tests for differences.**




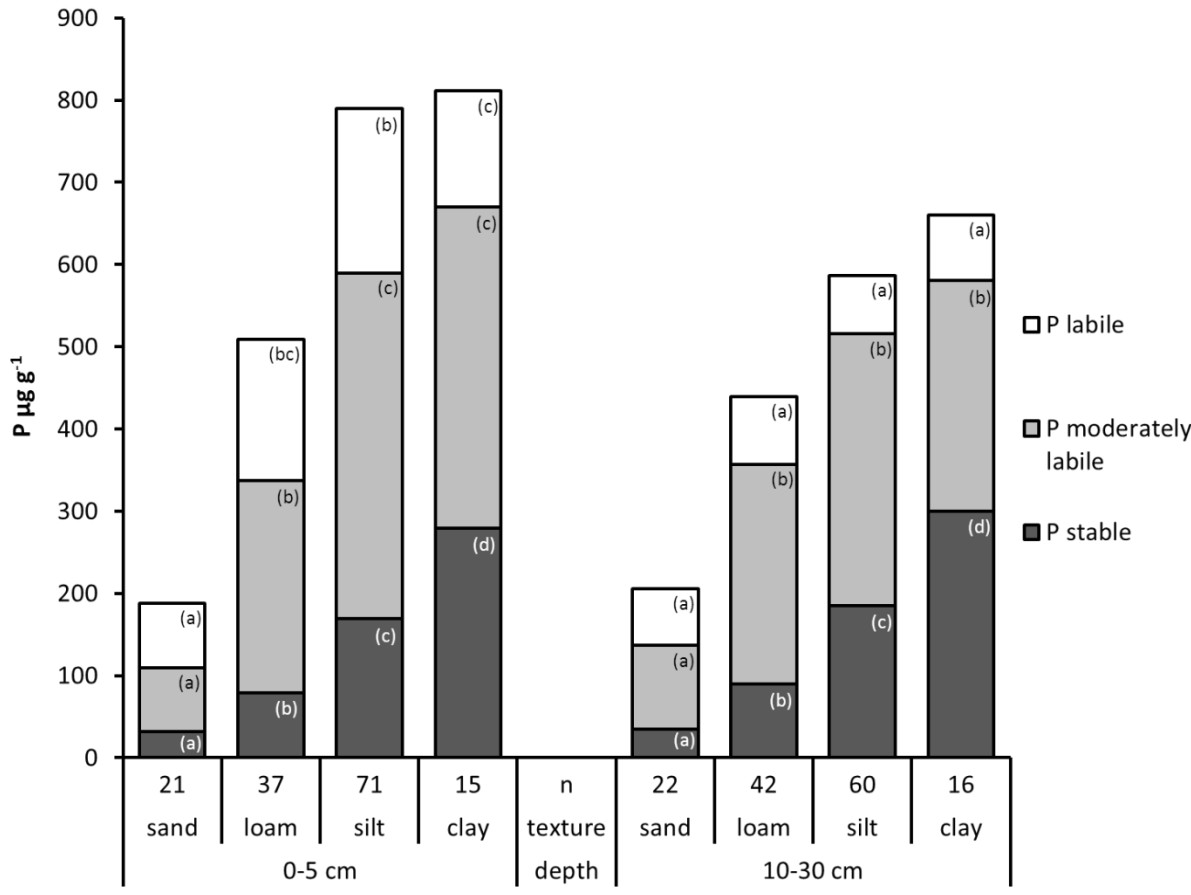

**Figure 5: Hedley P pools and total P (mean values) grouped by soil texture and depth, n = number of observations, lower case letters indicate significant differences in P pools among soil texture groups, non-parametric Mann-Whitney-U-test, α < 0.05.**



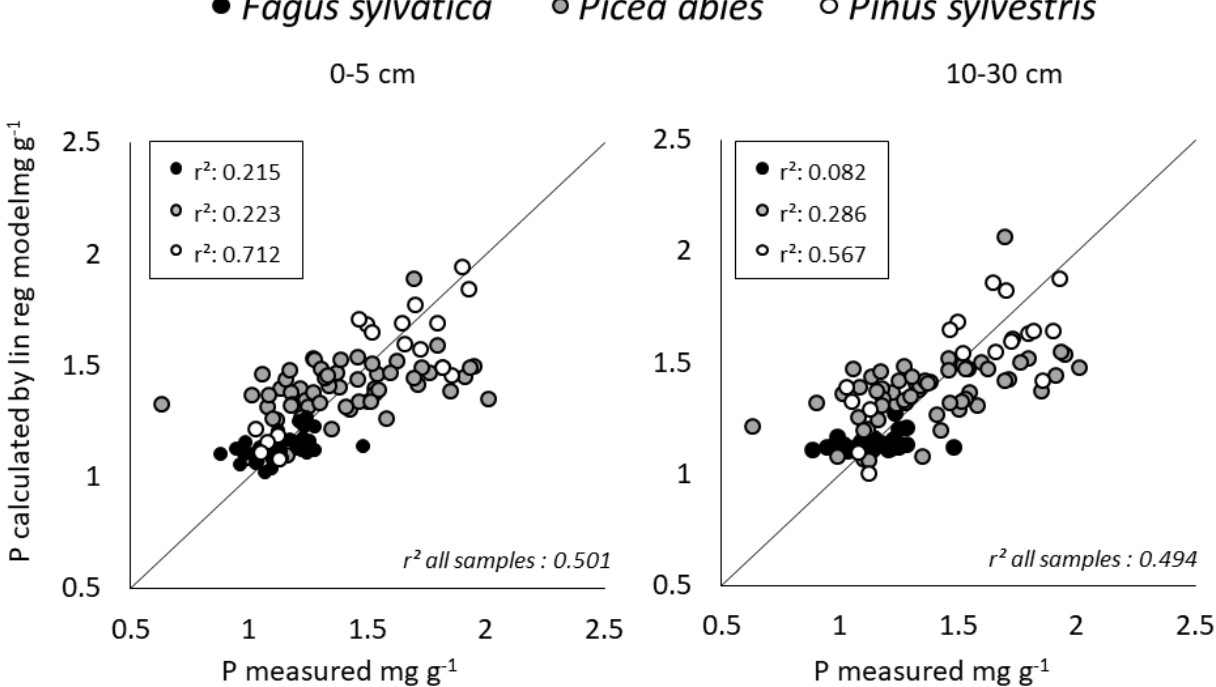

**Figure 6: Linear regressions to predict foliage P concentration in *P. abies*, *P. sylvestris*, and *F. sylvatica* (see Table 5) with soil variables from 0-5 (left panel) and from 10-30 cm depth (right panel), x-axis measured and y-axis modelled foliage P concentration in mg g⁻¹.**

