# Peer review of "Distribution of phosphorus fractions of different plant availability in German forest soils and their relationship to common soil properties and foliar P contents"

_SOIL, 2018_

## Referee Comment (RC1) · Anonymous Referee #1 · 2 Jan 2019

General comments:

The paper of Niederberger et al. al addresses a very important issue for both forest ecologists as well as forest managers, namely the identification of a suitable soil P (fraction) assessment which permits a direct precise judgment of the nutritional status of forest trees. This issue is currently getting ever more important, because many forest stands in Europe apparently are drifting from former N limitation into P limitation.

In the paper of Niederberger et al., the well-known Hedley P fractionation method is

applied for a large number of forest soil samples, which constitute a large portion of a nation-wide soil and forest nutrition inventory in Germany. The ample work allowed a test whether and with which precision the P nutritional status of important forest tree species types in Germany can be predicted by different Hedley fractions performed on mineral topsoil samples. To my knowledge such a large-scale assessment has been performed for the first time.

The main result of the paper is that – unfortunately – none of the different Hedley fractions was strongly correlated with the P nutritional status of forest trees as assessed by foliar analysis. To me this is not too surprising because both the forest floor as well as the subsoil which both are of critical importance for the P supply of forest trees particularly under conditions of poor P supply have not been included in the study. Nevertheless I feel that the study is an important step forward, in increasing our knowledge on interrelations between soil P fractions and tree nutrition – at least it convincingly shows that Hedley fractions obtained on mineral topsoil horizons enable a rough estimation of the P nutrition status of Pinus sylvestris, but are inappropriate to characterize the P nutrition of Picea abies and Fagus sylvatica. I therefore recommend publication of the paper after the concerns and issues raised below are addressed and the manuscript has been improved accordingly. Perhaps some of my recommendations might even result in an increased predictive power of the Hedley fractionation results.

1. The paper is rather long and sometimes cumbersome to read. It should be shortened by 20-30% to attract more readers and focused on the most relevant issues.

2. Soil texture and SOC at least in a given depth increment most often are strongly correlated with each other, because low sand and high (silt)/clay contents favor SOC accumulation by formation of organo-mineral associations and aggregates which impede SOC mineralization. This multicollinearity effect could be used either form an amalgamated predictor or to remove one of the two variables in order to shorten the paper.

3. The authors should test whether splitting the sample collective into 2 sub-collectives (non-calcareous soils [pH < 6.5] vs. calcareous soils [pH > 6.5]) may improve the predictive power of the Hedley fractions for characterizing the P nutritional status of the trees.

4. Moreover, I suggest to test whether the calculated total topsoil stocks of the different Hedley fractions (the latter should be available according to the statement made in page 12, line 19/20 of the manuscript) may improve the predictive power of the Hedley fractions for characterizing the P nutritional status of the trees.

5. There are several papers dealing with the relation between (operationally-defined) soil P fractions and the P nutritional status of German forest ecosystems, whose results could be compared with the results of the Hedley procedure. For example the papers of Prietzel and Stetter (2011), Prietzel et al. (2013; DOI 10.1007/s11104-014-2248-9); and the recent paper of Manghabati et al 2018 (JPNSS; DOI: 10.1002/jpln.201700536) all found a good predictive power of citric-acid extractable soil P on tree P nutrition, whereas $HCO_3$ was suitable only under particular conditions (Manghabati et al.).

6. At least in some pages, the paper contains a lot of typos/spelling mistakes and sloppy grammar. For example, page 10 reads

L2 "showed significant higher amount of organic P forms" L5 "influence of microorganism" L13 "These effect could also be observed" L13/14 "this increase (. . .) were (. . .) not significant"

The English should be brushed up before resubmission.

Specific Comments

P2 L34: Reference De Schrijver et al 2012 is missing in the Reference Section

P3 L10: C content. Is this total C or organic C? Should be clarified. The text on P4 L $\frac{3}{4}$ suggests that "C content" means SOC, whereas the large Max C/N values (437; 61) presented in Table 1 indicate that at in the calcareous soils C content included

inorganic C in addition to SOC.

P4 L9: In deciduous tree stands, leaves are always from the current year

P4 L22: Nitric acid digestion does not completely retrieve total P, because Si-bound P is only partially mobilized. Underestimation is between 15 and 37% (Schwartz & Kölbel, 1992; Z Pflanzenernähr Bodenkd 155: 281–284; Hornburg & Lüer 1999; J Plant Nutr Soil Sci 162:131–137.

P5 L3: Has the MWU-Test been corrected for multiple comparisons? Please indicate!

P13 L3: forest floor mass and Corg/Porg ratio or P , Porg content should be used

Figure 2: see comment to P4 L22

Figure 3: Nice figure.

Table 2: Pi residual: Is this really Pi or may it also include Porg which is liberated and converted into Pinorg by nitric acid/H2O2 digestion?

Table 2; Table 5: As the data are non-normally distributed would it make sense to describe the variation in box plots rather than by standard deviation, which requires normal distribution?

---

## Referee Comment (RC2) · Anonymous Referee #2 · 9 Jan 2019

The paper of Niederberger et al. is a valuable contribution to a differentiated understanding of the connections between soil P pools, soil properties and tree nutrition. Also none of the fractions was strongly correlated with the P nutrition the influence of the different soil properties onto the P content gives valuable insights. I recommend publication of the paper after some minor revisions.

Like in the comment 3 of Referee #1, I would also encourage the authors to try splitting the collective into non-calcareous and calcareous soils.

Some specific comments:

P4 L3: Total C or SOC?

P4 L10: Is it the most recent whorl? According to the BZE II manual by picea abies the 7th (to the 15th) whorl is recommended for needle analysis.

P4 L10: Needle and leaves were collected at the same time span (2006-2008 for GFSI II) not "at the same time"? For example sampling of beech leaves is not recommended in the autumn.

P12 L16: Or is it an effect of the soil texture since most of the P. sylvestris plots have sandy soils? Than there would be soil type-specific instead of species-specific differences.

P12 L16: P. abies, not Pi. abies

Table 1: Values for SOC should be included.

Table S6: Better SOC (under predictor variables) instead of Carbon (total?). In Table 3 it is called SOC.

In some pages are unnecessary hyphens in the text (for example: P1 L11, P1 L15, P10 L18&20, P11 L 12)

---

## Referee Comment (RC3) · Anonymous Referee #3 · 18 Jan 2019

The referee report is attached as pdf.

Please also note the supplement to this comment:
https://www.soil-discuss.net/soil-2018-40/soil-2018-40-RC3-supplement.pdf

---

## Author Comment (AC1) · 30 Mar 2019

The comment was uploaded in the form of a supplement:
https://www.soil-discuss.net/soil-2018-40/soil-2018-40-AC1-supplement.pdf

---

## Author Comment (AC2) · 30 Mar 2019

Here we listed our responses to the comments of reviewer 1 in tabular form. The page and line numbers of the referee's comments refer to the original manuscript: soil-2018-40, (https://doi.org/10.5194/soil-2018-40).
Page and line numbers of the Author's reply refer to the revised manuscript.
We want to thank the anonymous reviewer for the valuable input to improve the manuscript.
In behalf of all authors, Jörg Niederberger

| Index | Referee's comment | Author's reply |
|---|---|---|
| 1 | Like in the comment 3 of Referee #1, I would also encourage the authors to try splitting the collective into non-calcareous and calcareous soils. | Although we included a large number of sites in our survey, there were only 8 out of 143 sites with a soil pH above 6.5. This number is too low to develop robust statistical models for these calcareous soils. However, we checked also models excluding these 8 calcareous sites and compared them with models including all sites. Model results for the group of soil samples with pH < 6.5 (non-calcareous soils) did not change substantially when compared to models including all sites. We found only some minor improvements as well as some minor deterioration of model quality. Nevertheless, we could not observe changes in the selected predictor variables or in the dominant predictor variable for non-calcareous soils. See Methods section chapter 2.4, P5 L 13 ff. However, it would be very interesting to address the issue of calcareous soils in a future study with a different collective of soil samples. (See our response to Reviewer 1 comment 3) |
| P4 L3: | Total C or SOC? | We changed that to SOC |
| P4 L10: | Is it the most recent whorl? According to the BZE II manual by picea abies the 7th (to the 15th) whorl is recommended for needle analysis. | Indeed, we used the most recent needles from the 7th whorl. We clarified this in the text, Method section 2.1, P 4 L13 ff. |
| P4 L10: | Needle and leaves were collected at the same time span (2006-2008 for GFSI II) not "at the same time"? For example sampling of beech leaves is not recommended in the autumn. | Yes, leaf samples where not taken "at the same time" in the sense of a simultaneous sampling, but samples, that were used here, were taken in the same year. This has been corrected. |
| P12 L16: | Or is it an effect of the soil texture since most of the P. sylvestris plots have sandy soils? Than there would be soil type-specific instead of species-specific differences. | The differences are certainly not caused by tree species; it seems that we expressed this ambiguously. We rephrased the introductory paragraph of this chapter to clarify that soil parameters and P content were the driving factors of P supply of trees. |

| P12 L16: | P. abies, not Pi. abies | We changed this in the text. |
|---|---|---|
| Table S6: | Better SOC (under predictor variables) instead of Carbon (total?). In Table 3 it is called SOC. | We harmonized this and changed to SOC |
| | In some pages are unnecessary hyphens in the text (for example: P1 L11, P1 L15, P10 L18&20, P11 L 12) | We deleted unnecessary hyphens. |

---

## Author Comment (AC3) · 30 Mar 2019

The comment was uploaded in the form of a supplement:
https://www.soil-discuss.net/soil-2018-40/soil-2018-40-AC3-supplement.pdf

---

## Author Response (AR1)

Here we listed our responses to the comments of referee one to three in tabular form. We want to thank the three anonymous reviewers for the valuable input to improve the manuscript.

The page and line numbers of the referee's comments in the column "index" refer to the original manuscript: soil-2018-40, (https://doi.org/10.5194/soil-2018-40).
In our **Author's reply** we included the Section, chapter, page number and line number to ease identifying our changes in the revised manuscript. Therefore, page and line numbers in the column "Author's reply" in the document below refers to the revised manuscript.

In behalf of all authors, Jörg Niederberger

| Index | Comments of referee one | Author's reply |
|---|---|---|
| 1 | The paper is rather long and sometimes cumbersome to read. It should be shortened by 20-30% to attract more readers and focused on the most relevant issues. | All reviewers, including Reviewer 1 suggested a variety of improvements, which we implemented in the revised manuscript. These revisions should also improve the readability of the manuscript. At the same time, they did not allow for major reductions in the length of the paper. Furthermore, the other two reviewers did not criticize the length of the manuscript. Thus we believe that the size of our manuscript is appropriate to address all relevant issues of our study. |
| 2 | Soil texture and SOC at least in a given depth increment most often are strongly correlated with each other, because low sand and high (silt)/clay contents favor SOC accumulation by formation of organo-mineral associations and aggregates which impede SOC mineralization. This multicollinearity effect could be used either form an amalgamated predictor or to remove one of the two variables in order to shorten the paper. | We are aware of the problem of multicollinearity among predictor variables in soil. Therefore, we checked our predictor set before modeling for correlations, multicollinearity, as well as for autocorrelations. We could not observe any of the above mentioned effects. Additionally, the model output parameter provided by SPSS, as described in the material and method section (***Chapter 2.4. P 5 L 24 ff.***), provided no indication for autocorrelation (Durbin-Watson) or multicollinearity (VIF). Therefore, we believe that all predictors that we considered describe important and also different properties. |
| 3 | The authors should test whether splitting the sample collective into 2 sub-collectives (non-calcareous soils [pH < 6.5] vs. calcareous soils [pH > 6.5]) may improve the predictive power of the Hedley fractions for characterizing the P nutritional status of the trees. | Although we have a large number of sites included in our survey, there were only 8 out of 143 sites with a soil pH above 6.5. This number is too low to develop robust statistical models for these calcareous soils. However, we checked also models excluding these 8 calcareous sites and compared them with models including all sites. Model results for the group of soil samples with pH < 6.5 (non-calcareous soils) did not change substantially when compared to models including all sites. We found only some minor |

| | | improvements as well as some minor deterioration of model quality. Nevertheless, we could not observe changes in the selected predictor variables or in the dominant predictor variable for non-calcareous soils. See Methods section **chapter 2.4, P5 L 13 ff**.

However, it would be very interesting to address the issue of calcareous soils in a future study with a different collective of soil samples. |
|---|---|---|
| 4 | Moreover, I suggest to test whether the calculated total topsoil stocks of the different Hedley fractions (the latter should be available according to the statement made in page 12, line 19/20 of the manuscript) may improve the predictive power of the Hedley fractions for characterizing the P nutritional status of the trees. | We followed this suggestion and calculated also models with P stocks (stocks of total Hedley P and P Hedley P pools) as predictor variables to explain foliar P concentrations. However, we did not find any improvements in model quality, on the contrary, models were of consistently lower quality. One reason might be that we needed to make very broad assumptions for soil bulk density and stone content to calculate P stocks. . See Method section, *chapter 2.4 P5 L22ff*. |
| 5 | There are several papers dealing with the relation between (operationally-defined) soil P fractions and the P nutritional status of German forest ecosystems, whose results could be compared with the results of the Hedley procedure. For example the papers of Prietzel and Stetter (2011), Prietzel et al. (2013; DOI 10.1007/s11104-014-2248-9); and the recent paper of Manghabati et al 2018 (JPNSS; DOI: 10.1002/jpln.201700536) all found a good predictive power of citric-acid extractable soil P on tree P nutrition, whereas HCO3 was suitable only under particular conditions (Manghabati et al.). | We included these studies in the discussion section (**Chapter 4.2, P13 L10 ff., P14 L4 ff.**) of the revised manuscript. |
| 6 | At least in some pages, the paper contains a lot of typos/spelling mistakes and sloppy grammar. For example, page 10 reads. The English should be brushed up before resubmission | The paper has undergone a thorough language revision. |
| P2 L34 | Reference De Schrijver et al 2012 is missing in the Reference Section | The reference De Schrijver et al. 2012 was accidentally listed under „S". We corrected this in the revised manuscript. *(P 16, L1 ff)* |
| P3 L10: | C content. Is this total C or organic C? Should be clarified. The text on P4 L 34 suggests that "C content" means SOC, whereas the large Max C/N values (437; 61) presented in Table 1 indicate that at in the calcareous soils C | This was indeed SOC and therefore changed to soil organic carbon. Method section, *chapter 2.1, P4, L7* and in the following.
Unfortunately, the table in the manuscript was not the final version and included still samples from two sites with peaty soils. Here we found of |

| | content included inorganic C in addition to SOC. | course very high SOS values. These two sites were removed from our study. Table 1, *P21, L1 ff* |
|---|---|---|
| P4 L9: | In deciduous tree stands, leaves are always from the current year | We rephrased this sentence. Now it becomes clear that the leaves resp. needles were sampled in the same year as the soil samples were taken. Method section, ***P4, L13ff*** |
| P4 L22: | Nitric acid digestion does not completely retrieve total P, because Si-bound P is only partially mobilized. Underestimation is between 15 and 37% (Schwartz & Kölbel, 1992; Z Pflanzenernähr Bodenkd 155: 281–284; Hornburg & Lüer 1999; J Plant Nutr Soil Sci 162:131–137. | Indeed, that is the case; therefore, we used in the text usually the phrase "sum of all Hedley fractions". We think, that it is appropriate to use the same digestion technique like the strongest Hedley fractionation step to assess if the sum of the Hedley fractions corresponds with the independent P sum. If we had used the HF digestion instead, we were not able to compare the sum of fractions with the independently measured total P content. Nevertheless, we clarified this in the text, see Method section ***chapter 2.3, P4 L26ff***. |
| P5 L3: | Has the MWU-Test been corrected for multiple comparisons? Please indicate! | A correction of MWU-Test for multiple comparisons is necessary if multiple questions are tested simultaneously with one dataset. In our case, we tested our target soil variables independently with - according to the target variable - newly arranged datasets. Therefore, a Bonferroni correction was not necessary. |
| P13 L3: | forest floor mass and Corg/Porg ratio or P , Porg content should be used | We included this addition into the discussion section at this point, Discussion section, ***chapter 4.2, P13, L28-30.*** |
| Figure 2: | see comment to P4 L22 | See response to Comment P4 L22 |
| Figure 3: | Nice figure. | Thank you! |
| Table 2: | Pi residual: Is this really Pi or may it also include Porg which is liberated and converted into Pinorg by nitric acid/H2O2 digestion? | Indeed, digestions with strong acids could liberate organic P and convert it into inorganic P. We observed this problem in the fractionation step with concentrated HCl, where we found very inconsistent Po and Pi relations for repeatedly analysed samples, but very consistent results for the sum of Pi and Po. For that reason, we refrained from using Pi and Po for that fractionation step and instead, we used the sum of both. This issue was addressed in Niederberger et al. 2015 and 2016 and cited in our ***method section (P4, L18-19)***. Additionally, we didn't observe this phenomenon for the residual fraction in our preliminary studies, where we tried to address such problems. It might be that the organic P is already digested by the preceding $HCl_{conc}$ step which is also a strong acidic digestion. Nevertheless, we changed Pi residual into P residual to avoid any misunderstandings. |
| Table 2: Table 5: | As the data are non-normally distributed would it make sense to | We replaced the tables by box plots, Figure 3, ***P26, L1ff*** and Figure 7, **P30, L1ff.** |

| | describe the variation in box plots rather than by standard deviation, which requires normal distribution? | |
|---|---|---|
| **Index** | **Comments of referee two** | **Author's reply** |
| 1 | Like in the comment 3 of Referee #1, I would also encourage the authors to try splitting the collective into non-calcareous and calcareous soils. | Although we included a large number of sites in our survey, there were only 8 out of 143 sites with a soil pH above 6.5. This number is too low to develop robust statistical models for these calcareous soils. However, we checked also models excluding these 8 calcareous sites and compared them with models including all sites. Model results for the group of soil samples with pH < 6.5 (non-calcareous soils) did not change substantially when compared to models including all sites. We found only some minor improvements as well as some minor deterioration of model quality. Nevertheless, we could not observe changes in the selected predictor variables or in the dominant predictor variable for non-calcareous soils. See Methods section ***chapter 2.4, P5 L 13 ff***. However, it would be very interesting to address the issue of calcareous soils in a future study with a different collective of soil samples. (See our response to Reviewer 1 comment 3) |
| P4 L3: | Total C or SOC? | We changed that to SOC Method section, ***chapter 2.1, P4, L7*** and in the following. |
| P4 L10: | Is it the most recent whorl? According to the BZE II manual by picea abies the 7th (to the 15th) whorl is recommended for needle analysis. | Indeed, we used the most recent needles from the 7th whorl. We clarified this in the text, Method section, ***chapter 2.1, P 4 L13 ff***. |
| P4 L10: | Needle and leaves were collected at the same time span (2006-2008 for GFSI II) not "at the same time"? For example sampling of beech leaves is not recommended in the autumn. | Yes, leaf samples where not taken "at the same time" in the sense of a simultaneous sampling, but samples, that were used here, were taken in the same year. This has been corrected, Method section, ***chapter 2.1, P4, L12-14.*** |
| P12 L16: | Or is it an effect of the soil texture since most of the P. sylvestris plots have sandy soils? Than there would be soil type-specific instead of species-specific differences. | The differences are certainly not caused by tree species; it seems that we expressed this ambiguously. We rephrased the introductory paragraph of this chapter to clarify that soil parameters and P content were the driving factors of P supply of trees. Discussion section, *chapter 4.2, P12, L22ff* |
| P12 L16: | P. abies, not Pi. abies | We changed this in the text. |
| Table S6: | Better SOC (under predictor variables) instead of Carbon (total?). In Table 3 it is called SOC. | We harmonized this and changed to SOC, see author's reply to referee's comment P4 L3 |

| | In some pages are unnecessary hyphens in the text (for example: P1 L11, P1 L15, P10 L18&20, P11 L 12) | We deleted unnecessary hyphens. |
|---|---|---|
| **Index** | **Comments of Referee three** | **Author´s reply** |
| 1. | The English name and abbreviation for the inventory referring to in this study is "National Forest Soil Inventory in Germany (NFSI)". | We replaced GFSI by NFSI |
| 2. | Both "soil P (C, N) contents" and "soil P (C, N) concentrations" and both "foliar P contents" and "foliar P concentrations" have been used throughout the manuscript. Concentrations are defined as mass per volume (e.g., mg l-1); mass per mass (mg g-1) is called a content. Hence, please write "soil P (C, N) contents" and "foliar P contents" throughout the entire manuscript. | Changed and harmonized throughout the whole manuscript. |
| 3. | At some places expressions have been used that are – to my knowledge – not appropriate in the respective context or that have not been adequately explained/defined. For example P2 L8-9 "P cycling" and "intern reallocation (transfer) processes", P2 L14 "nutritional status", P3 L2 "population of inference", P10 L22 "distribution patterns", P12 L27 "distinct fractionation schemes". | We clarified and explained these expressions/concepts (one to three) or reworded them (four to six). Additionally we searched the manuscript thoroughly for ambiguous phrases. |
| 4. P2 L7: | Forest stands in Germany have partially been fertilized. Especially for stand establishment, fertilization including phosphorus has been a common measure in some regions. Additionally, phosphorus has been added in forest soil liming in some regions where total soil phosphorus pools are low. | We clarified this in the text. Introduction section, *P2, L6-9.* |
| 5. P2 L10-12: | Not only biomass harvesting is leading to nutrient deficiencies. Nitrogen input to forest ecosystems is also a driver for the establishment of nutrient deficiencies (e.g., | We included N deposition and soil acidification as examples of additional drivers of P nutrient deficiencies in the introduction section. Introduction section, *P2, L12-14.* |

| | | |
|---|---|---|
| | increased growth and therewith higher nutrient demand; changes in mycorrhizal symbioses; soil acidification). | |
| 6. P2 L14: | Define "nutritional status". From the following text it is obvious that foliar phosphorus contents are used as indicator for the nutritional status, but here it remains open. | We included the definition of "nutritional status" as foliar P content at this point in the text. Introduction section, *P2, L16-17.* |
| 7. P3 L8: | Which were the selection criteria for the subset? Why didn't you use all NFSI plots for which foliar phosphorus contents are available? | In our study, we needed to optimize the number of samples to keep the workload associated with the analysis manageable. Here the selection of sites and soil samples followed two distinct steps. Initially, the soils were selected to capture the variation in those properties that were relevant for the development of NIRS models. Specifically, we selected the NFSI soil samples to create NIRS models to predict P pools in mineral soils (compare: Niederberger, J., Todt, B., Boča, A., Nitschke, R., Kohler, M., Kühn, P. and Bauhus, J.: Use of near-infrared spectroscopy to assess phosphorus fractions of different plant availability in forest soils, Biogeosciences, 12, 3415–3428, doi:10.5194/bg-12-3415-2015, 2015). . Since it was not possible to analyze all existing NFSI plot, we tried to capture the major soil parent materials and different main tree species. The analyses of relationships between soil properties and foliage P content was a second step, that we had not foreseen when planning for the soil analyses. Thus we could only use all sites where we had performed a Hedley fractionation and where foliar data from the NFSI were available We clarified this in the introduction and material section. Introduction section, *P3, L11-13.* |
| 8. | Soil extraction methods indicative of the foliar P nutritional status are not only needed since the determination of foliar P contents is laborious and expensive, but also since foliar P contents have a large variability (among trees and among years). This large variability demands sampling of a large number of trees in several subsequent years in order to be able to evaluate the foliar P nutrition (Wehrmann 1959). Unfortunately, during NFSI only three trees in just one year have been sampled per plot. | Thank you for this suggestion, which we included in our discussion as one possible explanation of the weak coefficient of determination in the regression analysis. See *chapter 4.2 P12 L23 ff.* in the revised manuscript. |

| | | |
|---|---|---|
| | Hence, the NFSI dataset is on the one hand the largest forest soil dataset available in Germany, on the other hand foliar nutrient contents are afflicted with uncertainty due to the sampling design. Both the sampling design and the resulting uncertainty should be stated in the manuscript. This uncertainty in foliar phosphorus contents might be the reason for the small coefficient of determination in the regression analysis. | |
| 9. P3 L23: | In Table 1 the total P content is listed and in the abstract it is written that total P is commonly the only information on soil phosphorus in inventories; here you do not list the total P content as a parameter that was determined during the NFSI and on P4 L22-24 you describe the method used to determine total P. This is a bit confusing for the reader – did you determine total P by yourself or was the parameter provided by others? | Total P content was determined by the NFSI and we determined it in our study as the sum of our Hedley fractionation steps. In some cases we actually found substantial differences in total P contents (sum of all Hedley fractions) determined by us and provided in the NFSI data base. Therefore, we decided to include a nitric acid digestion, which was executed independently from the Hedley fractionation, to measure "total" P by ourselves
 We found a high level of agreement for our "total" P values and "total P" as the sum of all Hedley fractionation steps (r² 0.97). (We acknowledge that the nitric acid digestion does probably not extract all P; see our response to comment Reviewer 1, P4 L22).
 We clarified this in the Material and Method section in our revised manuscript, *chapter 2.3, P4, L26 ff.* |
| 10. P4 L9-10: | Beech trees just have current year leaves. Better write that the leaves were sampled from the upper crown. It is very uncommon that the most recent whorl is sampled. At least the NFSI samples taken by the Northwest German Research Institute were from the 7th to 12th whorl. | We clarified this in the text (compare response to comment of Reviewer 2 P4, L10)
 We used current year needles from the 7th whorl and clarified this point in the manuscript, *chapter 2.1, P4 L13 ff.* |
| 11. P8 L21-28: | What about the negative relationship between foliar P and SOC in the model for *F. sylvatica*? | The negative relationship between foliar P content and SOC was addressed in the discussion section (Chapter 4.2, P 13 L 24 ff. Nevertheless we emphasized this finding in the result section as well (*Chapter 3.7, P9 L7ff*). |
| 12. P9 L12-13: | Your results show that soil properties have an influence on Hedley P fractions and pools and that Hedley P fractions and pools do not explain the variance in foliar P contents very well. Hence, from your results, it is | This is a very valid point. We therefore addressed the issue of indication of plant availability of P in Hedley fractions in the discussion section *(P 12 L 30ff)* in the revised manuscript (see also response to comment of Reviewer 1 on P9 L12-13, orig. manuscript) |

| | questionable if Hedley P fractions represent plant available P fractions. | |
|---|---|---|
| 13.
P9 L30: | What do you mean with "within soil depth"? a) within one soil depth, b) within the
soil profile | Within the soil profile, we clarified this in the text. Discussion section, **chapter 4.1.1, P10, L14** |
| 14.
P10 L21: | Do you mean "DNA and phosphonate were only found in very acidic soils" or "DNA
and phosphonate were found in most acidic soils"? | Phosphonates were only found in acidic soils and the portion of DNA found in mineral soils increased with increasing acidity. We clarified this in the manuscript.
Discussion section, **chapter 4.1.2, P10 L32** |
| 15.
P10 L30-32: | Later on you discuss the effect of clay on P availability in detail. However, it is missing here, though it is necessary to understand your statement: Increased decomposition should increase labile P; however, many soils with high pH and large decomposition rates and intensive bioturbation probably have low sand/high clay contents leading to adsorption of P to clay minerals and therewith to small amounts of labile P. | We included a short explanation in the discussion **chapter 4.1.2, P11, L12-14**, to clarify our statement. |
| 16.
P10 L33-P11 L1: | Did you also include clay content instead of sand content in your regression analyses? | Yes, we did. Not surprisingly, the results showed opposite effects of the two predictors, since the increase in finer particles leads typically to a decrease in coarse particles and vice versa. Nevertheless we observed higher predictor strength for sand content than for clay content, therefore we decided to use sand content as a texture based predictor variable. |
| 17.
P11 L10-11: | Here and elsewhere you write about SOC, while in the material and methods section only the total C content is mentioned. Did you quantify carbonates in soils, too? Or did you exclude calcareous soils (seems not to be the case according to the pH values presented)? | We only determined SOC and thus have replaced "total C" with SOC throughout the manuscript, Method section, **chapter 2.1, P4, L7** and in the following, (see also response to comment 2 und 3 of Reviewer 1. |
| 18.
P11 L27-28: | Talkner et al. 2009 found a significant relationship between the clay content and organically bound P, too. | Yes, this was referenced in **P 12 L 4-6** (original manuscript) |
| 19.
P12 L6-8: | Where is this result shown (not in Table 5)? | The result is shown in Figure S3. We corrected this. Discussion section, **chapter 4.1.4, P12, L13.** |
| 20.
P12 L 26: | It was organic phosphorus (not carbon) and clay content that explained the variance in foliar P contents best. | Thank you for pointing this out. We corrected this in the manuscript. Discussion section, **chapter 4.2, P13, L18-19.** |

| 21.
P12 L33-
P13 L1: | Do you mean the negative relationship between SOC and foliar P content? | Yes, this has been modified. Discussion section, **chapter 4.2, P13, L16-29**. |
|---|---|---|
| 22.
P13 L13-15
and L19-
21: | Foliar P contents have a large variability (among trees and among years). This large variability demands sampling of a large number of trees in several
subsequent years in order to be able to evaluate the foliar P nutrition (Wehrmann 1959).
Unfortunately, during NFSI only three trees in just one year have been sampled per plot.
Hence, foliar nutrient contents are afflicted with uncertainty due to the sampling design. This
uncertainty in foliar phosphorus contents might be the reason for the small coefficient of
determination in the regression analyses. | See our response to referee threes comment 8 |

**Technical corrections referee three**

23. Different names have been used for the same thing. For example "foliage P contents" and "foliar P contents". Please harmonize the names.                                                    *done*

24. P3 L22: "North-West" -> "Northwest"                                                            *done*

25. P4 L28-29: "subject to" seems not to be the right word here.                                    *changed*

26. P5 L4 (and elsewhere): Better write "Hedley P pools", since the word "pools" is also used for masses related to an area (kg ha$_{-1}$).                                                      *changed*

27. At several places (e.g., P5 L4) hyphens occur in the middle of words.                          *deleted*

28. P5 L5: "Pools" probably has to be "P pools".                                                   *changed*

29. P5 L20 (and elsewhere): mg kg-1 -> mg kg$_{-1}$                                                 *done*

30. P8 L2: Delete the "and" at the end of the sentence.                                            *done*

31. P8 L23: "considerably" -> "considerable"                         *"varied considerably" (adverb!)*

32. P9 L1: (and elsewhere): "regressions models" -> "regression models"                            *done*

33. P9 L20: "org. C content" -> "organic C content"                                                *done*

34. P10 L5: "microorganism" -> "microorganisms"                                                    *changed*

35. P10 L13: "These effect" -> "This effect"                                                       *changed*

36. P10 L20: "even if there are" -> "even if there is"                                             *done*

37. P11 L32: "In forest soils of northern Germany" -> "In forest soils of northern and central Germany"                                                                                      *changed*

38. The bibliographical references are sometimes written with comma, sometimes without.
                                                                                                  *corrected*

39. P12 L5: "negative influence of P content in soils" -> "negative influence on P content in soils"
                                                                                                  *changed*

40. P12 L16: "Pi. abies" -> "P. abies"                                                             *changed*

41. P12 L23: "P fertilization lead to" -> "P fertilization leads to"      *changed to "P fertilization led to"*

42. P15 L18-19: The reference is incomplete.      *The reference is not incomplete, there is just a very unlucky formatting problem caused by a line break.*

43. P15 L33: "soils nutrients" -> "soil nutrients"      *changed*

44. P16 L1-2: The reference is incomplete.      *changed*

45. P26 Figure 2: "Po ready mineralizable" -> : "Po readily mineralizable" and "HNO3 65% +H2O2" -> "$HNO_3$ 65% + $H_2O_2$" and "grey boxes indicates" -> "grey boxes indicate" and "dashed line separates" -> "dashed lines separate"      *changed*

46. P28 Figure 4: "The column" -> "The columns"      *replaced*